# Psychometric properties of the Bangla version of the sense of coherence scale among university students in Bangladesh

**Momtaz Sultana[1], Yuta Hayashi**  **[2]\*, Tanzilur Rahman Tamim[3], Rie Chiba[4], Muhammad Kamal Uddin[1]**

1 Department of Psychology, University of Dhaka, Dhaka, Bangladesh, 2 Department of Nursing, Graduate School of Health Sciences, Kobe University, Kobe, Hyogo, Japan, 3 Department of Public Health, Graduate School of Health Sciences, Kobe University, Kobe, Hyogo, Japan, 4 Division of Psychiatric and Mental Health Nursing, Human Health Sciences, Graduate School of Medicine, Kyoto University, Kyoto, Japan

\* yuta-h@harbor.kobe-u.ac.jp

## Abstract

### Introduction

Sense of coherence is a key concept in psychological science that contributes to mental health by helping people cope with various stressors in their daily lives. A literature review demonstrated the unavailability of a tool for measuring sense of coherence in Bangladesh. This study aimed to translate the 13-item Sense of Coherence scale (SOC-13) into the Bangla language and to examine its validity, dimensionality, and reliability.

### Method

After translating the SOC-13, a cross-sectional online survey was conducted with 510 undergraduate students at a university in Bangladesh from July to October 2021. Construct validity was assessed through confirmatory factor analysis to analyze the structural validity of the Bangla SOC-13 scale, and by examining correlations with related constructs (self-esteem, well-being, and psychological distress). The alpha coefficient was calculated to examine internal consistency. Data were analyzed using SPSS version 22 and R statistical software.

### Results

A total of 320 respondents provided valid responses (response rate: 62.7%). The structural validity of the SOC-13, as examined through confirmatory factor analysis, was consistent across studies (a three-factor structure with an acceptable fit), and convergent validity was evidenced through a statistically significant positive relationships between sense of coherence and both self-esteem and well-being, as well as criterion validity was supported by a significant negative relationship with psychological distress. The internal consistency of the total scores by the coefficient alpha was good (α = 0.74), whereas the alphas of each subscale showed mediocre to fair reliability.

**Data availability statement:** All relevant data are within the paper and Supporting Information files.

**Funding:** This study was supported by the FY2021 research grant from the Gushin-kai (no grant number). The funder had no role in the acquisition or analysis of the data or in the content of this study.

**Competing interests:** The authors have declared that no competing interests exist.

## Conclusion

The Bangla version of the SOC-13 showed good construct validity, acceptable criterion validity, and good reliability based on overall internal consistency. Thus, it can be used to assess sense of coherence in young adults, although there is scope for further examination.

## Introduction

Studies on mental health have increasingly focused on the origins of health and well-being, through which individuals manage stress and stay healthy, rather than solely on the origins of the illness [1,2]. As significant personal resources in terms of health and well-being, generalized resistance resources (GRRs) and sense of coherence (SOC) are the core parts of salutogenesis [2]. Salutogenesis describes how humans deal with various stressors daily [3]. The term "generalized resistance resources" was coined by Antonovsky in 1979 and 1987 to describe the resources accessible to people, groups, or communities that help them cope with stressors and develop SOC [3–5]. A person can successfully manage stress using GRRs, which prevent tension induced by stressors from transforming into stress [6]. Through the accumulation of such experiences, an individual's SOC is formed and developed [5]. Strong SOC enables individuals to effectively identify and apply their GRRs in response to stressors [5]. It comprises three sub-components: comprehensibility, manageability, and meaningfulness [4,7]. Comprehensibility refers to the ability to perceive the world as consistent, predictable, and explicable. Manageability is defined as the ability to cope with daily demands and the perception of sufficient resources to deal with both internal and external stimuli. Meaningfulness refers to the idea that problems are sufficiently important to invest in and work on [4].

Researchers have found a strong correlation between SOC and mental health outcomes [6]. People with higher SOC levels are more likely to manage challenging circumstances effectively by using resources than those with lower SOC levels [8–10], leading to better mental and physical health and fulfilling lives [10–13]. People with high self-esteem are more likely to employ GRRs that reinforce SOC [14]. Strong SOC correlates with positive mental health and well-being [15,16] and functions as a robust predictor of health-promoting activities [17]. Conversely, an inverse relationship was observed between the SOC and psychological distress. In clinical populations, lower SOC scores have been linked to higher levels of psychological distress, suggesting that SOC may shield against mental health problems [18].

The SOC scale was developed by Antonovsky [4] to assess individualsP SOC levels, using an original 29-item version and a shortened 13-item version (SOC-13). SOC-13 has been translated and cross-culturally adapted [7,13,19]. The SOC-13 is a concise and easier-to-administer version of the SOC-29, resulting in higher response rates and better data quality. It retains good psychometric qualities, with reliability and validity comparable to the SOC-29 and has been demonstrated to be successful in a variety of settings [20]. Although some studies have eliminated one or two items to improve model fit [21–23], the three-factor structure is predominantly supported [7] with some exceptions [22,24–26].

In Bangladesh, a systematic review reported that the prevalence of mental illness ranged from 3.4% to 22.9% in children and from 6.5% to 31.0% in adults [27]. Despite the high prevalence and significant burden of mental illness, mental health services in Bangladesh are limited [28–30]. Addressing these issues is a critical challenge for the nation [31].

Young adults are particularly vulnerable to the negative effects of their socio-cultural environment [32–34]. They often exhibit behavioral changes such as unhealthy eating,

drinking, smoking, and sleeping problems [35–37], which can contribute to mental health issues [38,39]. As a result, public health policies are increasingly targeting young adults [39,40]. Since young adults constitute approximately 20% of the country's population [41], the psychological health of university students has recently garnered attention [42,43]. Focusing on improving SOC can help them manage stress more successfully, find more purpose in their lives, and enhance their overall mental health. This strategy not only meets urgent needs but also lays the groundwork for long-term mental health [2,4]. However, to our knowledge, there are no tools available to assess SOC in Bangladesh to date.

Therefore, this study aimed to translate the original English SOC-13 into Bangla and examine its validity and reliability among Bangladeshi university students. We hypothesized that the Bangla SOC-13 would show high internal consistency with a three-factor structure, as theorized. Based on earlier studies, we also hypothesized that SOC would be positively correlated with self-esteem and well-being and negatively correlated with psychological distress [12].

## Methods

### Translation process of the SOC-13 scale

After obtaining permission from the Society for Theory and Research on Salutogenesis (https://www.stars-society.org/) to translate the SOC-13 scale into Bangla, the instrument was adopted following the recommended procedures for scale translation [44].

First, two bilingual individuals (the first and third authors) independently translated the scale from the original English into Bangla. A discussion to merge the two translations was held among the research team members under the supervision of the fifth author. Second, a bilingual professional back-translated the scale into the English language. Subsequently, an expert (native English speaker) evaluated the original and back-translated versions of the scale. After detecting and handling translational differences based on expertise, the research team discussed and corrected them.

Cognitive interviews and pretests were conducted to check the feasibility and understandability of each item among 28 undergraduate students with different majors at a public university in Bangladesh. Participants were asked to respond to an online questionnaire on an individual basis. After revising the expressions of some items based on cognitive interviews, the Bangla version of the SOC-13 was completed.

### Participants and procedures

After the translation and finalization of the SOC-13, a cross-sectional survey was conducted with 510 students using an anonymous self-report online questionnaire who were selected through convenience sampling. The inclusion criteria were as follows: 1) undergraduate students at a university in Bangladesh, and 2) citizens and residents of Bangladesh.

According to the COSMIN guideline, a minimum of 10cases per item are required for confirmatory factor analysis [45]. For SOC-13, the minimum sample size requirement for conducting confirmatory factor analysis (CFA) was 130; however, a larger sample size is advisable. The participants were informed of the purpose and methods of the study and the access code for the online questionnaire was distributed. The questionnaire had a consent form to confirm consent to participate on the first page and was designed to end automatically if they did not agree. Responses were collected only from those who agreed to participate. The survey was conducted from 29th July to 31st October 2021. In compliance with the Declaration of Helsinki, this study was comprehensively approved by the Ethics Committee of Kobe University, Japan, to which the principal investigator (YH) was affiliated (No. 1020).

## Instruments

**SOC-13.** The SOC-13 contains three subcomponents: comprehensibility (items 2, 6, 8, 9, and 11), manageability (items 3, 5, 10, and 13), and meaningfulness (items 1, 4, 7, and 12) [4,46]. This scale is rated on a 7-point Likert-type scale with a possible range of scores between 13 and 91, with a higher score indicating a higher SOC level. The original version of the SOC-13 scale had a coefficient alpha that ranged from 0.70 to 0.92 [20].

**Rosenberg Self-Esteem Scale (RSES).** The RSES was originally developed to measure adolescents' feelings of self-worth and acceptance [47]. On this 10-item scale, the items were answered using a four-point response format as follows: 1 (Strongly disagree), 2 (Disagree), 3 (Agree), and 4 (Strongly agree). The scores on the scale range from 10 to 40, with higher scores representing higher self-esteem. Five items assessed positive feelings, and the remaining five assessed negative feelings. An example item of the scale is "On the whole, I am satisfied with myself." The Bangla RSES has shown high reliability (coefficient α = 0.87) and adequate concurrent validity [48].

**World Health Organization 5-Item Well-Being Index (WHO-5).** The WHO-5 is a generic global rating scale that measures subjective well-being [49]. It asks how well each of the five statements had been applied in the last two weeks. It is rated on a 6-point Likert-type scale: 0 (At no time), 1 (Some of the time), 2 (Less than half of the time), 3 (More than half of the time), 4 (Most of the time), and 5 (All the time). The scores range from 0 to 25, with higher scores representing higher subjective well-being. An example item of the WHO-5 index is "I have felt cheerful and in good spirits." Since its first publication in 1998, the WHO-5 has been translated into more than 30 languages and its psychometric properties have been established across different cultures with diverse samples [50]. The Bangla version of the WHO-5 has acceptable internal consistency, test-retest reliability, and convergent validity [51]. The WHO-5 in this study has shown high reliability (coefficient α = 0.89).

**Kessler psychological distress scale (K6).** The K6 is a simple measure of psychological distress [52]. Participants were asked to rate how often they felt in the past four weeks: (1) nervous, (2) hopeless, (3) restless or fidgety, (4) so depressed that nothing could help them, (5) that everything was an effort, and (6) worthless. Response options were 0 (None of the time), 1 (Little of the time), 2 (Some of the time), 3 (Most of the time), and 4 (All the time). An example item of the scale is "In the last four weeks, about how often did you feel hopeless?" The total K6 score ranged from 0 to 24, with higher scores indicating more severe psychological distress. The Bangla version of the K6 has acceptable internal consistency, test-retest reliability, and construct validity [53]. The K6 in this study has shown high reliability (coefficient α = 0.90).

## Data analysis

After excluding data with missing values on the SOC-13, demographic data were analyzed descriptively. The following analyses were performed according to the COSMIN guidelines [54,55] and construct validity was examined through CFA to assess the three-factor structures of comprehensibility, manageability, and meaningfulness. The fit indices included the chi-squared test, goodness-of-fit index (GFI), comparative fit index (CFI), Tucker-Lewis index (TLI), root mean square error of approximation (RMSEA), and standardized root mean square residual (SRMR). To assess model fit, we followed the popular suggestions of Hu and Bentler [56]: CFI and TLI: good fit ≥ 0.95, acceptable fit ≥ 0.90; RMSEA: good fit < 0.06, acceptable fit < 0.08; and SRMR: good fit < 0.08, acceptable fit < 0.10. When a good model fit was not observed for all 13 items (Model 1), the modified three-factor models were sequentially examined considering error variances and correlated residuals, in accordance with the

findings of previous studies [22,57,58]. In Model 2, we empirically allowed some pairs of items to covariate their error variances considering the similarity and relativity of these items.

After conducting the CFA, convergent and criterion validity were examined using correlations with self-esteem and well-being, and with psychological distress, respectively. The reliability of the total Bangladesh SOC-13 score was tested for internal consistency. We examined Cronbach's alpha as the reliability coefficient for the Bangla SOC-13 and its three subscales. SPSS version 22 (IBM, USA) was used for descriptive statistics, and R statistical software was used for CFA. Differences were considered statistically significant at P values of less than 0.05. All the tests were two-tailed.

## Results

### Sample description

A total of 510 undergraduate students in grades 1–4 from different faculty at a public university in Bangladesh met the inclusion criteria. Of these, 372 students participated in the study, and 52 incomplete responses were excluded. Consequently, 320 respondents provided valid responses (response rate: 62.7%). The participants' characteristics are presented in Table 1.

**Table 1. Demographic characteristics of the participants (*N* = 320).**

|  | N | (%) | Mean |
|---|---|---|---|
| **Gender** | | | |
| Male | 127 | 39.7 | |
| Female | 193 | 60.3 | |
| **Age** | | | 21.59 (SD = 1.51) |
| **Grade** | | | |
| 1st | 85 | 26.6 | |
| 2nd | 121 | 37.8 | |
| 3rd | 54 | 16.9 | |
| 4th | 59 | 18.4 | |
| Unknown | 1 | 0.3 | |
| **Major** | | | |
| Psychology | 225 | 70.3 | |
| Geography | 18 | 5.6 | |
| Sociology | 10 | 3.1 | |
| Political Science | 10 | 3.1 | |
| Others | 54 | 16.9 | |
| Unknown | 3 | 0.9 | |
| **Housing** | | | |
| Parents' home | 252 | 78.8 | |
| Dormitory | 41 | 12.8 | |
| Apartment | 20 | 6.3 | |
| Others | 5 | 1.6 | |
| Unknown | 2 | 0.6 | |
| **Partner** | | | |
| Yes | 70 | 21.9 | |
| No | 250 | 78.1 | |

## Mean and standard deviation of the Bangla SOC-13

Table 2 presents the mean values and standard deviations for each item. None of the items exhibited ceiling or floor effects.

## Construct validity of the Bangla SOC-13

**Structural validity.** The statistical properties of the two models obtained using CFA are listed in Table 3. First, the CFA with all 13 items (Model 1) resulted in a poor fit (CFI = 0.83, TLI = 0.78, RMSEA = 0.10, and SRMR = 0.09). Subsequently, Model 2 with the modification indices as four error covariances was tested, i.e., item 2 "Has it happened in the past that you were surprised by the behavior of people whom you thought you knew well? (1 = Never happened, to 7 = Always happened)" (comprehensibility dimension) and item 3 "Has it happened that people whom you counted on disappointed you? (1 = Never happened, to 7 = Always happened)" (manageability dimension), item 2 and item 4 "Until now your life has had: (1 = No clear goals or purpose at all, to 7 = Very clear goals and purpose)" (meaningfulness dimension), item 2 and item 10 "Many people—even those with a strong character—sometimes feel like sad sacks (losers) in certain situations. How often did you feel this way in the past? (1 = never, 7 = very often)" (manageability dimension), and items 4 and 10. This yielded an acceptable model fit (CFI = 0.93, TLI = 0.91, RMSEA = 0.06, and SRMR = 0.07). Model 2 (Fig 1) revealed standardized item loadings across three latent constructs:

**Table 2. Mean and standard deviation of each item of the Bangla SOC-13 (*N* = 320).**

| Item no. | Item description | Mean | SD |
|---|---|---|---|
| 1 | Do you have the feeling that you don't really care about what goes on around you? | 4.4 | 2.1 |
| 2 | Has it happened in the past that you were surprised by the behavior of people whom you thought you knew well? | 2.9 | 1.7 |
| 3 | Has it happened that people whom you counted on disappointed you? | 3.5 | 1.8 |
| 4 | Until now your life has had: | 4.7 | 2.1 |
| 5 | Do you have the feeling that you're being treated unfairly? | 4.2 | 2.2 |
| 6 | Do you have the feeling that you are in an unfamiliar situation and don't know what to do? | 3.7 | 2.3 |
| 7 | Doing the things, you do every day is: | 4.5 | 1.7 |
| 8 | Do you have very mixed-up feelings and ideas? | 3.2 | 1.9 |
| 9 | Does it happen that you have feelings inside you would rather not feel? | 3.4 | 2.2 |
| 10 | Many people – even those with a strong character – sometimes feel like sad sacks (losers) in certain situations. How often have you felt this way in the past? | 3.5 | 2.0 |
| 11 | When something happened, have you generally found that: | 5.2 | 1.5 |
| 12 | How often do you have the feeling that there's little meaning in the things you do in your daily life? | 4.0 | 2.2 |
| 13 | How often do you have feelings that you're not sure you can keep under control? | 3.7 | 2.2 |

SOC-13, the 13-item version of the sense of coherence scale.

**Table 3. Comparison of three-factor analysis models in the Bangla SOC-13 (*N* = 320).**

| | Chi-square | df | *p* | GFI | CFI | TLI | RMSEA | SRMR |
|---|---|---|---|---|---|---|---|---|
| Model 1 (13 items) | 242.93 | 62.00 | < 0.01 | 0.99 | 0.83 | 0.78 | 0.10 | 0.09 |
| **Model 2 (13 items)** **(Modified model with four modification indices)** | **126.81** | **58.00** | **< 0.01** | **0.99** | **0.93** | **0.91** | **0.06** | **0.07** |

SOC-13, 13-item version of the sense of coherence scale; df, degrees of freedom; GFI, goodness-of-fit index; CFI, comparative fit index (desired ≥ 0.90); TLI, Tucker-Lewis index (desired ≥ 0.90); RMSEA, root mean square error of approximation (desired < 0.08); SRMR, standardized root mean square residual (desired < 0.10).

Note. Best fitting model in bold.

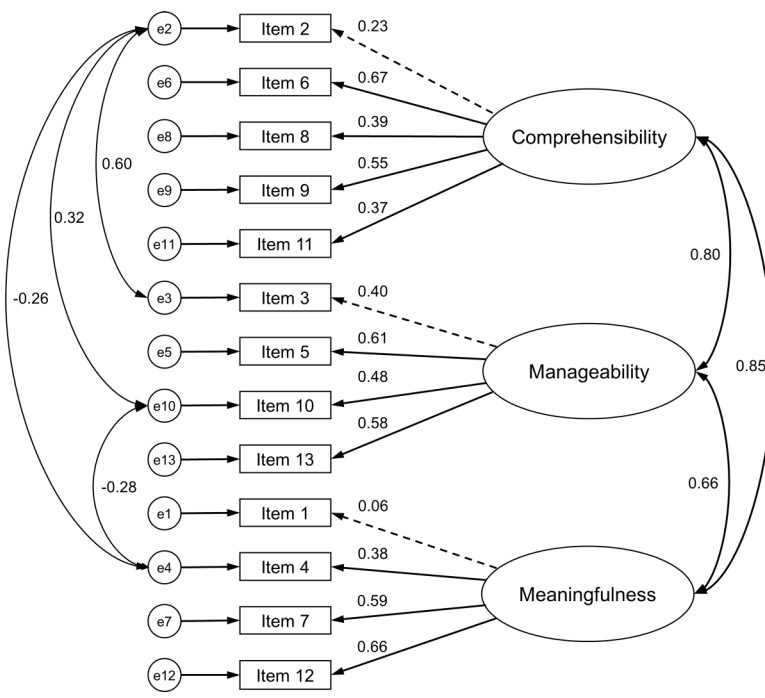

**Fig 1. Model 2 with four modification indices.**

comprehensibility, meaningfulness, and manageability. This model indicated significant item loadings across most components, although items 1 (0.06), 2 (0.23), and 3 (0.40) had weak loadings in the dimensions of meaningfulness, comprehensibility, and manageability, respectively indicating a poor fit for these items. Model 2 also revealed the following correlation coefficients: 0.80 between comprehensibility and manageability, 0.85 between comprehensibility and meaningfulness, and 0.66 between manageability and meaningfulness (Fig 1). Model 2 showed a better fit than Model 1. Although the factor loadings of items 1 and 2 were fairly weak (0.06 and 0.23, respectively), Model 2 was regarded as the final model in this study.

**Convergent and criterion validity.** SOC-13 showed statistically significant positive correlations with RSES (r = 0.58; $p < 0.01$) and WHO-5 (r = 0.46; $p < 0.01$), and a statistically significant negative correlation with K6 (r = − 0.63; $p < 0.01$).

## Reliability of the Bangla version of SOC-13

The alpha coefficients for the comprehensibility, manageability, and meaningfulness subscales were 0.56, 0.60, and 0.41, respectively, indicating mediocre-to-fair reliability. The reliability coefficient of the total SOC-13 score is 0.74, indicating good internal consistency.

## Discussion

This study translated the SOC 13-item scale from English to Bangla, and evaluated its psychometric properties among university students in Bangladesh. Good construct validity including acceptable structural validity, and overall reliability were found good, while the subscale reliability was fair. Interestingly, our findings revealed significant error covariances between the four item pairs across the three SOC dimensions, implying that these items are interconnected and may not work independently.

The current study demonstrated the Bangla SOC-13 scale's three-factor structure. However, error covariances were found between certain item pairings (2 and 3, 2 and 4, 2 and 10, and 4 and 10) belonging to comprehensibility, manageability, and meaningfulness. Items 2 and 3 address people's expectations of whom they can rely and their discrepancies. These components overlapped thematically, explaining the observed error covariance. The second pair of items, 2 and 4, included perspectives related to unexpected situations. In terms of the third pair of items, items 2 and 10, and the fourth pair of items, items 4 and 10, refer to how to perceive things and hold perspectives. Similar empirical covariances have been previously reported [22,57,58]. For example, the three-factor model of the Slovenian SOC-13 scale was modified to allow for correlated residuals for two pairs of items (items 2 and 3, as well as items 4 and 13) [58]. In addition, the Ethiopian SOC-13 scale showed relatively weak item loadings ($\beta < 0.40$) for Items 1, 2, 3, and 12 [22].

The acceptance of these error covariances is supported by both theoretical and empirical evidence. Antonovsky observed that SOC aspects do not always recognized clearly, resulting in overlaps [46]. Despite this, maintaining the three-factor model is critical because it accords with Antonovsky's original idea of SOC, which emphasizes the multidimensional aspects of resilience and coping. Furthermore, incorporating these error covariances significantly enhances the model fit, as shown in prior studies with lower RMSEA and better CFI values [21,59]. Empirical evidence from various populations has consistently supported this three-factor structure, demonstrating its validity and efficacy in capturing the complexity of the SOC construct [21,23]. Consequently, despite certain overlapping of measurements, the three-factor approach ensures both theoretical and statistical integrity in the SOC assessment. Given the empirical overlap between SOC-13 elements from multiple dimensions [4,20], earlier research has suggested analyzing the total SOC-13 score. Based on the good overall reliability and fair subscale reliability, we suggest that the Bangla SOC-13 can be considered for use as a total score.

Our results also showed that SOC had a significant positive correlation with self-esteem and well-being, and a significant negative correlation with psychological distress. These findings are consistent with those of previous studies [12,60]. SOC may have a strong effect on subjective health, including mental health, and well-being [6,61]. A systematic review of adolescents revealed that higher SOC levels indicate lower stress and lower internalized or externalized difficulties, as SOC can be viewed as a resilience factor [62]. Consequently, the Bangla version of the SOC-13 scale used in this study appears to be a measure for demonstrating good construct and criterion validity.

The reliability coefficient for the total SOC-13 score was satisfactory, as observed in previous studies [23,63]. The mediocre to fair coefficient alphas of the subscales, as observed in earlier studies [21,58], can be attributed to the small number of scale items. Further verification including test-retest reliability is required in future studies.

This study has certain limitations. First, the survey was conducted at only one university and most participants majored in psychology. Further studies involving participants from different backgrounds, including various academic specialties and residential settings, may produce more robust and compelling results. Second, cross-cultural applicability was not rigorously examined. Although the translation process was skillfully conducted, equivalence across cultures in terms of language, beliefs, and context may still have scope for further examination. Third, data collection was based on a self-reported online questionnaire; therefore, social desirability bias might have existed, even though it was completely anonymous. Despite these limitations, the Bangla version of the SOC-13 will be utilized in future studies to reveal the factors associated with SOC and to obtain several suggestions on how SOC can be maintained and enhanced in Bangladesh.

## Conclusions

The Bangla SOC-13 is an appropriate SOC measure for Bangladesh. The scale demonstrated good construct validity, including acceptable structural and convergent validity; acceptable criterion validity; good overall reliability based on internal consistency. Future research with more diverse and larger samples should be conducted to rigorously verify and establish psychometric properties. Despite these limitations, the findings have significant implications for psychology and public health research in Bangladesh. The validated Bangla SOC-13 can be a useful instrument for future SOC studies and strategies to enhance well-being.

## Supporting information

**S1 Table. Spreadsheet file of dataset.**
(XLSX)

**S1 File. Inclusivity in global research questionnaire.**
(DOCX)

## Acknowledgments

We appreciate Ms. Naima Kayser Leeoza (Karolinska Institutet) and Dr. Glen David Edwards (Kobe University) for their contribution to the translation process; Dr. Mushfiqul Anwar Siraji (North South University) for statistical advice.

## Author contributions

**Conceptualization:** Momtaz Sultana, Yuta Hayashi, Tanzilur Rahman Tamim, Rie Chiba, Muhammad Kamal Uddin.

**Data curation:** Momtaz Sultana, Tanzilur Rahman Tamim.

**Formal analysis:** Momtaz Sultana.

**Funding acquisition:** Yuta Hayashi.

**Investigation:** Momtaz Sultana, Tanzilur Rahman Tamim.

**Project administration:** Yuta Hayashi.

**Supervision:** Yuta Hayashi, Rie Chiba, Muhammad Kamal Uddin.

**Visualization:** Momtaz Sultana.

**Writing – original draft:** Momtaz Sultana, Tanzilur Rahman Tamim.

**Writing – review & editing:** Yuta Hayashi, Rie Chiba, Muhammad Kamal Uddin.

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
