## [Decision Letter · Decision Letter 0]

29 Aug 2024

PONE-D-24-25645Psychometric Properties of the Bangla Version of the Sense of Coherence Scale among University Students in BangladeshPLOS ONE

Dear Dr. Hayashi,

Thank you for submitting your manuscript to PLOS ONE. After careful consideration, we feel that it has merit but does not fully meet PLOS ONE’s publication criteria as it currently stands. Therefore, we invite you to submit a revised version of the manuscript that addresses the points raised during the review process.

We look forward to receiving your revised manuscript.

Kind regards,

Taro Matsuki, Ph.D.

Guest Editor

PLOS ONE

Journal Requirements:

3. We note that there is identifying data in the Supporting Information file <S1_Table >. Due to the inclusion of these potentially identifying data, we have removed this file from your file inventory. Prior to sharing human research participant data, authors should consult with an ethics committee to ensure data are shared in accordance with participant consent and all applicable local laws.

-Location data

Reviewers' comments:

Reviewer's Responses to Questions

**Comments to the Author**

1. Is the manuscript technically sound, and do the data support the conclusions?

Reviewer #1: Partly

Reviewer #2: Partly

2. Has the statistical analysis been performed appropriately and rigorously? 

Reviewer #1: Yes

Reviewer #2: Yes

3. Have the authors made all data underlying the findings in their manuscript fully available?

Reviewer #1: No

Reviewer #2: Yes

4. Is the manuscript presented in an intelligible fashion and written in standard English?

Reviewer #1: Yes

Reviewer #2: No

5. Review Comments to the Author

Reviewer #1: In order to maintain and improve mental health, it is very important to understand the health aspects of the subject. Therefore, creating a scale that captures sense of coherence would be an important research topic.

However, there are some unclear sections in it, and we cannot recommend publication it at present. My comments are shown below, so please deal specifically with each one.

[Major comments]

1. (Introduction) Please provide a detailed review of the relationship between the three variables used as indicators of construct validity (self-esteem, well-being, and psychological distress) and SOC, and then state your hypothesis. For example, what kind of relationships have been shown in previous studies?

This information is necessary to determine whether each indicator is appropriate as an indicator of construct validity.

2. (Instruments) Please use a consistent description for [Instruments].

Please indicate the internal consistency coefficient for K6 and WHO-5, and for K6, explain what higher scores mean. Also, please include demographic information (what was listened to and how).

3. (Results) Please provide descriptive statistics for each measure.

In SOC-13, correlations between each factor are reported as [internal consistency reliability], which I do not think is suitable as an indicator of reliability. In CFA, correlations between each factor are calculated, which can cope with the issue of attenuation of correlation coefficients (i.e., values close to the true correlation coefficient are calculated). I think it would be better to report the correlation coefficients between factors based on the CFA results in the chapter [Factorial validity]. Please consider this.

[Minor comments]

1. Translation process of the SOC-13 scale (p.5, l.101~)

MS, TRT, and MKU refer to authors?

It is unclear what they stand for, so please add a note or state "first author" etc.

2.Sample description and Table1

Please make sure the statistics in Table 1 are consistent with those in the text (gender percentage). The total number of students in the grade is 319. Is there one unknown? If that's so, please add it.

３. Figure1

Please provide an explanation for the numbers and arrows in Figure 1.

Reviewer #2: The authors have attempted to determine the psychometric properties of the Bangla version of the SOC-13 scale. The findings will contribute to the existing literature on the measurement of SOC by adding a version of the SOC-13 in a different language and context. However, several aspects of the manuscript need to be refined to present the findings more scientifically and enhance readability. The authors are encouraged to consider the following suggestions and revise their manuscript accordingly:

General Comments:

• The authors should use consistent terminology throughout the manuscript. Factor analysis is a method for establishing construct validity, while testing against similar and dissimilar constructs is also a method of establishing construct validity. In some literature, this is referred to as testing criterion validity. It appears that the authors have assessed convergent and discriminant validity by examining correlations with other constructs, such as self-esteem, psychological distress, and mental well-being. One could also argue that criterion validity has been established. Therefore, it would be best for the authors to clarify their intentions and ensure the readers have a clear understanding. Please revise the methods section in the abstract, the aim in the introduction (page 5, lines 89–90), the data analysis in the methods section (page 9, line 182), and the relevant sections in the discussion and conclusion accordingly.

• Please include keywords in the abstract that use Medical Subject Headings (MeSH) to facilitate indexing and searching in MEDLINE/PubMed and other databases.

• The authors should address grammatical errors and improve the English writing to enhance readability and comprehension.

Introduction:

• Lines 55-56: The health model of salutogenesis involves a complex interplay between two key concepts: Sense of Coherence (SOC) and Generalized Resistance Resources (GRRs). The authors should provide readers with information about SOC, including its interaction with GRRs.

• Lines 58-59: SOC is defined as the ability to appropriately utilize GRRs to promote one's own health. The statement in the mentioned lines should be rephrased for better clarity.

• Lines 62-66: Please break down this sentence into shorter, more meaningful sentences to improve readability.

• Line 72: The statement could be rephrased as, "SOC-13 has been translated and cross-culturally adapted." As noted in the literature, the SOC scale has been tested for its validity and reliability in various contexts, not merely translated.

• Lines 76-88: The third paragraph of the introduction should be rephrased and realigned. Clearly justify how the health model of salutogenesis and a tool to measure SOC can be utilized to promote mental health and support efforts to mitigate the current mental disease burden in Bangladesh.

• Lines 77-79: The authors mention limitations in human resources for mental health and health services separately. These two are undoubtedly interrelated, so please rephrase for better clarity.

• Line 89: It would be helpful if the authors justified in the second and third paragraphs of the introduction why they decided to adapt the shorter version of SOC-13 rather than the original SOC-29.

Methods:

• Line 104: Did the authors mean to say "back-translated into English"? Please rephrase for clarity.

• Line 116: Please provide information about the sample size. How was the sample size determined, and what sampling method was used?

• Line 182: It appears that the authors have tested for convergent and discriminant validity. Did you use a multitrait-multimethod matrix for this analysis? The analysis could also be interpreted as testing for criterion validity. The authors should use precise terminology and provide detailed descriptions of the analysis methods to avoid any ambiguity for the readers.

Results:

• Line 200: In Table 1, it appears the row corresponding to "Females" is missing. Please ensure this is included.

• Line 203: Ensure consistent terminology throughout the manuscript. Ensure that the most appropriate term is used

• Table 2: It is unclear what the authors aim to convey by reporting item means and standard deviations. While it is statistically valid to report these, what do they signify in the context of the study? According to Antonovsky’s original concept, SOC is considered a single construct, and he was critical of reporting scores under three different domains due to his belief in its unified nature. Although modern statistical techniques have led to such analyses in the literature, interpreting single-item statistics may not provide meaningful insights for this construct. Therefore, this Table may be redundant. The authors may consider to replace it with a Table that includes factor loadings for each item, which would be more informative. However, authors may accept or decline this suggestion based on valid reasons.

• Table 3: It would be best to add footnotes indicating the cut-off levels you considered for a good fit. For example, "CFI – Comparative Fit Index (desired >0.9)."

• Line 241: While "test-retest reliability" is a commonly used term, "internal consistency reliability" might not be as widely recognized. Consider revising the sub-title to "Reliability of the Bangla Version of SOC-13" and simply provide the relevant values. Additionally, in the methods section (lines 184-185), clearly state that reliability was measured through internal consistency.

Discussion:

In general, the discussion should be strengthened to provide a meaningful analysis based on the current study's findings. Therefore, I suggest the authors rewrite the discussion section using the following sub-themes:

• Summary of principal findings: Provide a concise summary without reiterating the results.

• Strengths and weaknesses of the study: Discuss the key strengths and any limitations encountered during the study.

• Comparison with existing evidence: Explain how the current study's findings compare or contrast with the existing literature. This should include evidence that both supports and contradicts the study's findings. When discussing similarities or differences with previous research, base the arguments on the strengths and weaknesses of the methodologies and contexts of the studies rather than personal opinions.

• Implications for future research: Discuss the potential impact of the findings on future research directions.

Specific Comments:

• Lines 255-256: I suggest that the authors provide a brief summary of the findings here. What do the authors mean by "good construct validity"?

• Second Paragraph, Line 260: The authors mention that they considered certain pairs of items to share similar concepts during the analysis. What evidence led to this conclusion? According to Table 4, there is a weak correlation between manageability and meaningfulness in the study population. The authors paired items 4 and 10, assuming these two items share similar constructs. It would be beneficial to justify this analysis and provide evidence for and against these methods from the literature.

Additionally, since these items, which were modified to share covariances, belong to different domains, does this also suggest that, even though the study identified a three-factor model, there is some interchange of items between different domains? There is existing literature on modified SOC scales that supports this idea. It is well established that there is a correlation between different domains and each domain with the total SOC score, as the authors have mentioned. Have the authors considered assessing a one-factor model? Several studies have proposed a one-factor model with a good fit, which aligns with Antonovsky’s original hypothesis. Regional studies from India and Sri Lanka among young cohorts have also suggested such findings. A more in-depth discussion on this aspect would be valuable.

• Lines 271-272: The authors suggest using the SOC score as a total score. What specific findings of this study support this recommendation? Please elaborate.

• Lines 283-284: The statement, "Therefore, the results of this study are sufficient to demonstrate high construct validity," is unclear. Is this conclusion referring to the original SOC scale or the Bangla version? This statement needs to be rephrased for clarity.

• Lines 290-291: The authors claim that the sample size of the present study was inadequate and consider this a weakness. However, based on the literature on sample size determination for factor analysis, a sample size of 320 is generally considered adequate for a questionnaire with only 13 items.

• Limitations: One of the limitations of this study could be that the authors did not investigate item and conceptual validity (conceptual equivalence). Conceptual equivalence assesses whether the constructs and domains in the original instrument are equally relevant and appropriate in the target culture. This is an important step that researchers often overlook during the adaptation of study instruments.

• Lines 295-296: The intended meaning of this statement is unclear. Please rephrase for clarity.

• Lines 298-302: This section appears to be repeated in the conclusion section. The authors can omit this section.

Conclusions:

The conclusion needs to be rephrased to provide a proper summary of the study findings using appropriate terms.

6. PLOS authors have the option to publish the peer review history of their article (what does this mean? ). If published, this will include your full peer review and any attached files.

**Do you want your identity to be public for this peer review?** For information about this choice, including consent withdrawal, please see our Privacy Policy .

Reviewer #1: No

Reviewer #2: No

---

## [Author Response · Author response to Decision Letter 0]

25 Oct 2024

RESPONSE TO REVIEWERS

Dear Editor and Reviewers,

We apologize for the delay in submitting the revised manuscript. We appreciate the time and effort by the editor and reviewers in considering this manuscript and allowing us to submit a revised version to PLOS ONE. We are grateful to the editor and reviewers for the constructive feedback and helpful suggestions.

We have revised the manuscript in accordance with the review comments. Our responses are presented in detail as follows, as well as a revised version of the manuscript with changes highlighted in red.

Taking into account the contributions to this study, including the revisions, the order of authors has been rearranged. All authors have agreed to this change.

Response to Comments from Reviewer #1

Comment #1-1:(Introduction) Please provide a detailed review of the relationship between the three variables used as indicators of construct validity (self-esteem, well-being, and psychological distress) and SOC, and then state your hypothesis. For example, what kind of relationships have been shown in previous studies? This information is necessary to determine whether each indicator is appropriate as an indicator of construct validity.

Response:

Thank you for your valuable comments. The relationship between the three variables has been added with references (lines 74-80) as follows: “People with high self-esteem are more likely to employ GRRs that reinforce SOC. Strong SOC correlates with positive mental health and well-being and functions as a robust predictor of health-promoting activities. Conversely, an inverse relationship was observed between the SOC and psychological distress. In clinical populations, lower SOC scores have been linked to higher levels of psychological distress, suggesting that SOC may shield against mental health problems”.

Comment #1-2:(Instruments) Please use a consistent description for [Instruments].

Please indicate the internal consistency coefficient for K6 and WHO-5, and for K6, explain what higher scores mean. Also, please include demographic information (what was listened to and how).

Response:

Thank you for the comments. We added the mentioned descriptions of the instruments. Please check in the instruments section.

Comment #1-3:(Results) Please provide descriptive statistics for each measure.

In SOC-13, correlations between each factor are reported as [internal consistency reliability], which I do not think is suitable as an indicator of reliability. In CFA, correlations between each factor are calculated, which can cope with the issue of attenuation of correlation coefficients (i.e., values close to the true correlation coefficient are calculated). I think it would be better to report the correlation coefficients between factors based on the CFA results in the chapter [Factorial validity]. Please consider this.

Response:

The correlation coefficients between factors based on CFA results has been added in the structural validity section (lines: 252-254).

Comment #1-4: Translation process of the SOC-13 scale (p.5, l.101~)

MS, TRT, and MKU refer to authors? It is unclear what they stand for, so please add a note or state "first author" etc.

Response:

We apologize for the confusing description. We have revised it to state that “first author” etc., respectively (lines 118-121).

Comment #1-5: Sample description and Table1

Please make sure the statistics in Table 1 are consistent with those in the text (gender percentage). The total number of students in the grade is 319. Is there one unknown? If that's so, please add it.

Response:

Thank you for noticing it. We revised according to the suggestion (one participant didn’t respond).

Comment #1-6: Figure1

Please provide an explanation for the numbers and arrows in Figure 1.

Response:

Thanks for the valuable comments. Explanation was added (lines 247-252) as follows: “Model 2 (Fig1) revealed standardized item loadings across three latent constructs: comprehensibility, meaningfulness, and manageability. This model indicated significant item loadings across most components, although items 1 (0.06), 2 (0.23), and 3 (0.40) had weak loadings in the dimensions of meaningfulness, comprehensibility, and manageability, respectively indicating a poor fit for these items”.

Response to Comments from Reviewer #2

Comment #2-1: The authors should use consistent terminology throughout the manuscript. Factor analysis is a method for establishing construct validity, while testing against similar and dissimilar constructs is also a method of establishing construct validity. In some literature, this is referred to as testing criterion validity. It appears that the authors have assessed convergent and discriminant validity by examining correlations with other constructs, such as self-esteem, psychological distress, and mental well-being. One could also argue that criterion validity has been established. Therefore, it would be best for the authors to clarify their intentions and ensure the readers have a clear understanding. Please revise the methods section in the abstract, the aim in the introduction (page 5, lines 89–90), the data analysis in the methods section (page 9, line 182), and the relevant sections in the discussion and conclusion accordingly.

Response:

We appreciate your comments. We revised the mentioned sections according to the COSMIN guideline. Please check the abstract (lines 31-33), introduction (lines107-108), and data analysis (lines 207-210) sections. Furthermore, although the analysis could be interpreted as testing criterion validity, our primary aim was to assess construct validity, focusing specifically on structural and convergent validity. We have clarified this in the revised methods section to ensure precise terminology and avoid ambiguity for the readers.

Comment #2-2: Please include keywords in the abstract that use Medical Subject Headings (MeSH) to facilitate indexing and searching in MEDLINE/PubMed and other databases.

Response:

Thank you for the helpful suggestion. Keywords in the abstract using Medical Subject Headings was added (Bangladesh; Psychometrics; Scale; Sense of coherence; University students).

Comment #2-3: The authors should address grammatical errors and improve the English writing to enhance readability and comprehension.

Response:

The revised manuscript has been proofread again to avoid grammatical errors and enhance readability.

Introduction:

Comment #2-4: Lines 55-56: The health model of salutogenesis involves a complex interplay between two key concepts: Sense of Coherence (SOC) and Generalized Resistance Resources (GRRs). The authors should provide readers with information about SOC, including its interaction with GRRs.

Response:

We revised the introduction section based on the reviewer’s helpful suggestion. The interaction between SOC and GRRs as core concepts of salutogenesis has been added in the introduction section (lines 55-64) as follows: “As significant personal resources in terms of health and well-being, generalized resistance resources (GRRs) and sense of coherence (SOC) are the core parts of salutogenesis. Salutogenesis describes how humans deal with various stressors daily. The term "generalized resistance resources" was coined by Antonovsky in 1979 and 1987 to describe the resources accessible to people, groups, or communities that help them cope with stressors and develop SOC. A person can successfully manage stress using GRRs, which prevent tension induced by stressors from transforming into stress. Through the accumulation of such experiences, an individual's SOC is formed and developed. Strong SOC enables individuals to effectively identify and apply their GRRs in response to stressors”.

Comment #2-5: Lines 58-59: SOC is defined as the ability to appropriately utilize GRRs to promote one's own health. The statement in the mentioned lines should be rephrased for better clarity.

Response:

We have rephrased the sentences (lines 63-64) as follows: “Strong SOC enables individuals to effectively identify and apply their GRRs in response to stressors”.

Comment #2-6: Lines 62-66: Please break down this sentence into shorter, more meaningful sentences to improve readability.

Response:

Thanks for the suggestion. We revise the sentences as per the comments (lines 66-70).

Comment #2-7: Line 72: The statement could be rephrased as, "SOC-13 has been translated and cross-culturally adapted." As noted in the literature, the SOC scale has been tested for its validity and reliability in various contexts, not merely translated.

Response:

We rephrased the sentence (lines 82-83), “SOC-13 has been translated and cross-culturally adapted”.

Comment #2-8: Lines 76-88: The third paragraph of the introduction should be rephrased and realigned. Clearly justify how the health model of salutogenesis and a tool to measure SOC can be utilized to promote mental health and support efforts to mitigate the current mental disease burden in Bangladesh.

Response:

This Paragraph was rephrased and realigned according to the comment (lines 90-105).

Comment #2-9: Lines 77-79: The authors mention limitations in human resources for mental health and health services separately. These two are undoubtedly interrelated, so please rephrase for better clarity.

Response:

We have rephrased this sentence (lines 91-93) as, “Despite the high prevalence and significant burden of mental illness, mental health services in Bangladesh are limited”.

Comment #2-10: Line 89: It would be helpful if the authors justified in the second and third paragraphs of the introduction why they decided to adapt the shorter version of SOC-13 rather than the original SOC-29.

Response:

Thank you for the valuable suggestion. We have added the justification for using the shorter version of SOC scale (lines 82-87) as follows: “The SOC-13 is a concise and easier-to-administer version of the SOC-29, resulting in higher response rates and better data quality. It retains good psychometric qualities, with reliability and validity comparable to the SOC-29 and has been demonstrated to be successful in a variety of settings”.

Methods:

Comment #2-11: Line 104: Did the authors mean to say "back-translated into English"? Please rephrase for clarity.

Response:

We have rephrased the sentence for clarity (line 121).

Comment #2-12: Line 116: Please provide information about the sample size. How was the sample size determined, and what sampling method was used?

Response:

We appreciate your comments. Information about the sample was added in the participants and procedures of methods section (lines 134, 137-140). We used convenience sampling technique for data collection and for sample size determination we followed COSMIN guidelines.

Comment #2-13: Line 182: It appears that the authors have tested for convergent and discriminant validity. Did you use a multitrait-multimethod matrix for this analysis? The analysis could also be interpreted as testing for criterion validity. The authors should use precise terminology and provide detailed descriptions of the analysis methods to avoid any ambiguity for the readers.

Response:

We did not employ a multitrait-multimethod matrix in this study, as our analysis was based on a single method of measurement (Pearson’s correlation coefficients) and aimed to test relationships between SOC-13 and other constructs. We followed the COSMIN guidelines, which support correlation analysis as an appropriate method for construct validity when using a single method.

Results:

Comment #2-14: Line 200: In Table 1, it appears the row corresponding to "Females" is missing. Please ensure this is included.

Response:

Thank you for mentioning it. We have added the row corresponding to "Female" in Table 1.

Comment #2-15: Line 203: Ensure consistent terminology throughout the manuscript. Ensure that the most appropriate term is used.

Response:

We ensured that the key terms were used consistently across all sections, including the introduction, methods, results, and discussion. Specifically: The term "structural validity" has been consistently used instead of the previously interchangeable term "factorial validity". We have removed any instances of ambiguity or inconsistency in terminology related to construct validity to ensure clarity.

Comment #2-16: Table 2: It is unclear what the authors aim to convey by reporting item means and standard deviations. While it is statistically valid to report these, what do they signify in the context of the study? According to Antonovsky’s original concept, SOC is considered a single construct, and he was critical of reporting scores under three different domains due to his belief in its unified nature. Although modern statistical techniques have led to such analyses in the literature, interpreting single-item statistics may not provide meaningful insights for this construct. Therefore, this Table may be redundant. The authors may consider to replace it with a Table that includes factor loadings for each item, which would be more informative. However, authors may accept or decline this suggestion based on valid reasons.

Response:

Table 2 presents the mean values and standard deviations for each item to see whether any item exhibited a ceiling or floor effect. On the other hand, it is incongruous to arrange the items by domain before presenting the results of the factor analysis, so the items were sorted by item number.

Comment #2-17: Table 3: It would be best to add footnotes indicating the cut-off levels you considered for a good fit. For example, "CFI – Comparative Fit Index (desired >0.9)."

Response:

Thank you for the suggestion. Footnote was added according to the comment (lines 260-263) as follows: “df, degrees of freedom; GFI, goodness-of-fit index; CFI, comparative fit index (desired ≥ 0.90); TLI, Tucker-Lewis index (desired ≥ 0.90); RMSEA, root mean square error of approximation (desired < 0.08); SRMR, standardized root mean square residual (desired < 0.10)”.

Comment #2-18: Line 241: While "test-retest reliability" is a commonly used term, "internal consistency reliability" might not be as widely recognized. Consider revising the sub-title to "Reliability of the Bangla Version of SOC-13" and simply provide the relevant values. Additionally, in the methods section (lines 184-185), clearly state that reliability was measured through internal consistency.

Response:

Thank you for the reviewer’s suggestion. We have revised the subheading to "Reliability of the Bangla Version of SOC-13" (line 273) to use a more widely recognized term. Additionally, we clarified in the methods section (lines 207-210) that reliability was measured through internal consistency, using Cronbach’s alpha as recommended by the COSMIN guidelines.

Discussion:

Comment #2-19: In general, the discussion should be strengthened to provide a meaningful analysis based on the current study's findings. Therefore, I suggest the authors rewrite the discussion section using the following sub-themes:

• Summary of principal findings: Provide a concise summary without reiterating the results.

• Strengths and weaknesses of the study: Discuss the key strengths and any limitations encountered during the study.

• Comparison with existing evidence: Explain how the current study's findings compare or contrast with the existing literature. This should include evidence that both supports and contradicts the study's findings. When discussing similarities or differences with previous research, base the arguments on the strengths and

---

## [Decision Letter · Decision Letter 1]

18 Nov 2024

PONE-D-24-25645R1Psychometric properties of the Bangla version of the sense of coherence scale among university students in BangladeshPLOS ONE

Dear Dr. Hayashi,

Thank you for submitting your manuscript to PLOS ONE. After careful consideration, we feel that it has merit but does not fully meet PLOS ONE’s publication criteria as it currently stands. Therefore, we invite you to submit a revised version of the manuscript that addresses the points raised during the review process.

Please address the following comments to ensure the manuscript's quality and acceptability:

Line 106- 108; “Therefore, this study aimed to translate the original English SOC-13 into Bangla and examine its construct validity and reliability based on its internal consistency among Bangladeshi university students

Comment: The sentence above implies that both construct validity and reliability were assessed using internal consistency. To improve clarity, the authors should rephrase this. Additionally, since the latter part of the manuscript specifies that both structural and convergent validity were tested, it would be more accurate to state that "validity and reliability" were assessed at this stage. Now that the methods section includes detailed descriptions of the types of validity measurements conducted, this will provide clearer information for the readers.

Line 138-140- The authors' sample size calculation and their decision to include a larger sample size are not aligned. It is rather speculative than scientifically convincing. Both smaller and larger sample sizes than calculated have limitations regarding methodological rigor and ethical considerations.

Line 207-208; convergent validity was examined using correlations with self-esteem, well-being, and psychological distress

Comment: Isn’t it discriminant validity that the authors wish to test between SOC and psychological distress?

Line 329-330- ‘Second, there was a lack of assessment of the items and cross-cultural validity’.

I assume that authors meant to say that items and concept of the questionnaire was not tested for its’ conceptual validity. Would suggest the authors to rephrase according to their intention.

Line 340-341—what about the convergent and discriminant validity? The study has shown significant positive correlation between SOC and similar construct as well as significant negative correlations between SOC and dissimilar construct. Perhaps the authors can state that in the conclusion. 

Authors should discuss the study's implications in the conclusion section, considering its originality.

During the revision process, please use track changes or a distinct color for text modifications.The article's language must adhere to the journal's standards.

We look forward to receiving your revised manuscript.

Kind regards,

Md. Fouad Hossain Sarker, MSS

Academic Editor

PLOS ONE

Journal Requirements:

Additional Editor Comments:

Dear Dr. Yuta Hayashi

I am delighted to inform you that the review report from the reviewers has been received. Reviewer 1 accepted your manuscript, while Reviewer 2 provided comments for further development. Please address the following comments to ensure the manuscript's quality and acceptability: 

Line 106- 108; “Therefore, this study aimed to translate the original English SOC-13 into Bangla and examine its construct validity and reliability based on its internal consistency among Bangladeshi university students

Comment: The sentence above implies that both construct validity and reliability were assessed using internal consistency. To improve clarity, the authors should rephrase this. Additionally, since the latter part of the manuscript specifies that both structural and convergent validity were tested, it would be more accurate to state that "validity and reliability" were assessed at this stage. Now that the methods section includes detailed descriptions of the types of validity measurements conducted, this will provide clearer information for the readers.

Line 138-140- The authors' sample size calculation and their decision to include a larger sample size are not aligned. It is rather speculative than scientifically convincing. Both smaller and larger sample sizes than calculated have limitations regarding methodological rigor and ethical considerations.

Line 207-208; convergent validity was examined using correlations with self-esteem, well-being, and psychological distress

Comment: Isn’t it discriminant validity that the authors wish to test between SOC and psychological distress?

Line 329-330- ‘Second, there was a lack of assessment of the items and cross-cultural validity’.

I assume that authors meant to say that items and concept of the questionnaire was not tested for its’ conceptual validity. Would suggest the authors to rephrase according to their intention.

Line 340-341- what about the convergent and discriminant validity? The study has shown significant positive correlation between SOC and similar construct as well as significant negative correlations between SOC and dissimilar construct. Perhaps the authors can state that in the conclusion.

Reviewers' comments:

Reviewer's Responses to Questions

**Comments to the Author**

1. If the authors have adequately addressed your comments raised in a previous round of review and you feel that this manuscript is now acceptable for publication, you may indicate that here to bypass the “Comments to the Author” section, enter your conflict of interest statement in the “Confidential to Editor” section, and submit your "Accept" recommendation.

Reviewer #1: All comments have been addressed

Reviewer #2: (No Response)

2. Is the manuscript technically sound, and do the data support the conclusions?

Reviewer #1: Yes

Reviewer #2: Partly

3. Has the statistical analysis been performed appropriately and rigorously? 

Reviewer #1: Yes

Reviewer #2: Yes

4. Have the authors made all data underlying the findings in their manuscript fully available?

Reviewer #1: Yes

Reviewer #2: Yes

5. Is the manuscript presented in an intelligible fashion and written in standard English?

Reviewer #1: Yes

Reviewer #2: Yes

6. Review Comments to the Author

Reviewer #1: We have reviewed the contents.

Thank you for addressing the feedback provided throughout the review process.

Reviewer #2: The authors have significantly improved the manuscript by addressing each of the reviewers' comments. However, a few minor revisions are still needed to enhance the clarity and quality of the manuscript. The authors are encouraged to consider the following suggestions and revise their manuscript accordingly,

Line 106- 108; “Therefore, this study aimed to translate the original English SOC-13 into Bangla and examine its construct validity and reliability based on its internal consistency among Bangladeshi university students

Comment; The sentence above implies that both construct validity and reliability were assessed using internal consistency. To improve clarity, the authors should rephrase this. Additionally, since the latter part of the manuscript specifies that both structural and convergent validity were tested, it would be more accurate to state that "validity and reliability" were assessed at this stage. Now that the methods section includes detailed descriptions of the types of validity measurements conducted, this will provide clearer information for the readers.

Line 138-140- The authors' sample size calculation and their decision to include a larger sample size are not aligned. It is rather speculative than scientifically convincing. Both smaller and larger sample sizes than calculated have limitations regarding methodological rigor and ethical considerations.

Line 207-208; convergent validity was examined using correlations with self-esteem, well-being, and psychological distress

Comment: Isn’t it discriminant validity that the authors wish to test between SOC and psychological distress?

Line 329-330- ‘Second, there was a lack of assessment of the items and cross-cultural validity’.

I assume that authors meant to say that items and concept of the questionnaire was not tested for its’ conceptual validity. Would suggest the authors to rephrase according to their intention.

Line 340-341- what about the convergent and discriminant validity? The study has shown significant positive correlation between SOC and similar construct as well as significant negative correlations between SOC and dissimilar construct. Perhaps the authors can state that in the conclusion.

7. PLOS authors have the option to publish the peer review history of their article (what does this mean? ). If published, this will include your full peer review and any attached files.

**Do you want your identity to be public for this peer review?** For information about this choice, including consent withdrawal, please see our Privacy Policy .

Reviewer #1: No

Reviewer #2: No

---

## [Author Response · Author response to Decision Letter 1]

31 Dec 2024

Dear Editor and Reviewers,

We sincerely appreciate the time and effort you have dedicated to reviewing our manuscript. We are deeply grateful to the editor and reviewers for your insightful comments and valuable feedback, which have greatly contributed to improving the quality of our work.

We have revised the manuscript in accordance with the review comments. Our responses are presented in detail as follows, as well as a revised version of the manuscript with track changes. We have also reviewed our reference list and removed Ilyas’s literature because it was found that the original text is currently unable to be read. The removal of this paper will not change the content of our text (Line 166-167).

We have re-read the revised manuscript whether the article's language adhere to the journal's standards. If there are any obvious linguistic or grammatical errors, please let us know.

Response to Comments from Reviewer #2

Comment #2-1:

Line 106- 108; “Therefore, this study aimed to translate the original English SOC-13 into Bangla and examine its construct validity and reliability based on its internal consistency among Bangladeshi university students

Comment: The sentence above implies that both construct validity and reliability were assessed using internal consistency. To improve clarity, the authors should rephrase this. Additionally, since the latter part of the manuscript specifies that both structural and convergent validity were tested, it would be more accurate to state that "validity and reliability" were assessed at this stage. Now that the methods section includes detailed descriptions of the types of validity measurements conducted, this will provide clearer information for the readers.

Response:

Thank you for your insightful comment. We have revised the sentence to improve clarity (Line 109-110): “Therefore, this study aimed to translate the original English SOC-13 into Bangla and examine its validity and reliability among Bangladeshi university students.”

Comment #2-2:

Line 138-140; The authors' sample size calculation and their decision to include a larger sample size are not aligned. It is rather speculative than scientifically convincing. Both smaller and larger sample sizes than calculated have limitations regarding methodological rigor and ethical considerations.

Response:

Thank you for providing helpful feedback on our sample size estimate and decision to incorporate a larger sample. We would like to clarify our approach. Based on the COSMIN guidelines, we determined that the minimum sample size for the SOC-13 scale is 130 cases, or 10 case each item. This figure ensured that we met the fundamental prerequisites for performing confirmatory factor analysis (CFA). However, we chose a larger sample size to strengthen our results, addressing any missing data. Considering larger sample sizes are commonly used in psychometric research to obtain robust results, we hope it is still acceptable.

Comment #2-3:

Line 207-208; convergent validity was examined using correlations with self-esteem, well-being, and psychological distress

Comment: Isn’t it discriminant validity that the authors wish to test between SOC and psychological distress?

Response:

Thank you very much for your insightful comment. We apologize for the insufficient response to your previous comments (#2-1 and #2-13), despite having received similar feedback before. As you pointed out, sense of coherence and psychological distress are distinct concepts, and thus it is not appropriate to examine convergent validity based on their relationship. Our intention in employing K6 for assessing psychological distress in this study was to investigate the criterion validity of the Bangla SOC-13 because previous studies reported a high negative correlation between them. To present this, we revised the manuscript as follows (Line 209-211): “...convergent and criterion validity were examined using correlations with self-esteem and well-being, and with psychological distress, respectively.” We have also revised the relevant description in the manuscript in line with this modification (Line 41-43, Line 269, Line 321).

Comment #2-4:

Line 329-330- ‘Second, there was a lack of assessment of the items and cross-cultural validity’.

I assume that authors meant to say that items and concept of the questionnaire was not tested for its’ conceptual validity. Would suggest the authors to rephrase according to their intention.

Response:

Thank you for the feedback. We revised the article to address the issues of conceptual validity and cross-cultural applicability. We have added the following line to the limitations section (Line 329-330): “Second, cross-cultural applicability was not rigorously examined.”

Comment #2-5:

Line 340-341- what about the convergent and discriminant validity? The study has shown significant positive correlation between SOC and similar construct as well as significant negative correlations between SOC and dissimilar construct. Perhaps the authors can state that in the conclusion.

Response:

Based on the results regarding the validity, we have added the following description to the conclusion section (Lines 340-343): “The scale demonstrated good construct validity, including acceptable structural and convergent validity; acceptable criterion validity; good overall reliability based on internal consistency." We have also revised the relevant description in the abstract (Line 47-48).

Comment #2-6:

Authors should discuss the study's implications in the conclusion section, considering its originality.

Response:

Thank you for your comment. We have described the study's implications as follows (Lines 344-347): “Despite these limitations, the findings have significant implications for psychology and public health research in Bangladesh. The validated Bangla SOC-13 can be a useful instrument for future SOC studies and strategies to enhance well-being."

---

## [Decision Letter · Decision Letter 2]

27 Feb 2025

Psychometric properties of the Bangla version of the sense of coherence scale among university students in Bangladesh

PONE-D-24-25645R2

Dear Dr. Yuta Hayashi 

We’re pleased to inform you that your manuscript has been judged scientifically suitable for publication and will be formally accepted for publication once it meets all outstanding technical requirements.

Kind regards,

Maria José Nogueira, Ph.D.

Academic Editor

PLOS ONE

Additional Editor Comments (optional):

Dear Authors

All concerns and suggestions for improving the manuscript were provided.

Therefore, the manuscript is ready for publication.

Reviewers' comments:

Reviewer's Responses to Questions

**Comments to the Author**

1. If the authors have adequately addressed your comments raised in a previous round of review and you feel that this manuscript is now acceptable for publication, you may indicate that here to bypass the “Comments to the Author” section, enter your conflict of interest statement in the “Confidential to Editor” section, and submit your "Accept" recommendation.

Reviewer #1: All comments have been addressed

Reviewer #2: All comments have been addressed

2. Is the manuscript technically sound, and do the data support the conclusions?

Reviewer #1: Yes

Reviewer #2: Yes

3. Has the statistical analysis been performed appropriately and rigorously? 

Reviewer #1: Yes

Reviewer #2: Yes

4. Have the authors made all data underlying the findings in their manuscript fully available?

Reviewer #1: Yes

Reviewer #2: Yes

5. Is the manuscript presented in an intelligible fashion and written in standard English?

Reviewer #1: Yes

Reviewer #2: Yes

6. Review Comments to the Author

Reviewer #1: (No Response)

Reviewer #2: (No Response)

7. PLOS authors have the option to publish the peer review history of their article (what does this mean? ). If published, this will include your full peer review and any attached files.

**Do you want your identity to be public for this peer review?** For information about this choice, including consent withdrawal, please see our Privacy Policy .

Reviewer #1: No

Reviewer #2: No

---

## [Editor Report · Acceptance letter]

PONE-D-24-25645R2

PLOS ONE

Dear Dr. Hayashi,

I'm pleased to inform you that your manuscript has been deemed suitable for publication in PLOS ONE. Congratulations! Your manuscript is now being handed over to our production team.

Kind regards,

on behalf of

Professor Maria José Nogueira

Academic Editor

PLOS ONE